# Mesenchymal Stem Cells in Radiation-Induced Pulmonary Fibrosis: Future Prospects

**DOI:** 10.3390/cells12010006

**Published:** 2022-12-20

**Authors:** Yusha Chen, Xuefeng Liu, Zhaohui Tong

**Affiliations:** 1Department of Respiratory and Critical Care Medicine, Beijing Institute of Respiratory Medicine, Beijing Chao-Yang Hospital, Capital Medical University, Beijing 100020, China; 2Departments of Pathology, Urology, and Radiation Oncology, College of Medicine, OSU Comprehensive Cancer Center, The Ohio State University, Columbus, OH 43210, USA

**Keywords:** mesenchymal stem cells, radiation-induced pulmonary fibrosis, exosomes, therapy

## Abstract

Radiation-induced pulmonary fibrosis (RIPF) is a general and fatal side effect of radiotherapy, while the pathogenesis has not been entirely understood yet. By now, there is still no effective clinical intervention available for treatment of RIPF. Recent studies revealed mesenchymal stromal cells (MSCs) as a promising therapy treatment due to their homing and differentiation ability, paracrine effects, immunomodulatory effects, and MSCs-derived exosomes. Nevertheless, problems and challenges in applying MSCs still need to be taken seriously. Herein, we reviewed the mechanisms and challenges in the applications of MSCs in treating RIPF.

## 1. Introduction

### 1.1. Radiation-Induced Pulmonary Fibrosis (RIPF)

RIPF is a common and serious adverse effect of radiotherapy for thoracic malignancies, including lung, breast, and esophageal cancer. The average incidence of RIPF varies, ranging from 16% to 28% [1]. RIPF is considered to be the chronic injury stage of radiation-induced lung injury(RILI), which often occurs 1 year after radiotherapy, while the acute injury stage—radiation pneumonitis (RP)—mostly occurs within 12 weeks [2]. It manifests as progressive dyspnea, irreversible destruction of lung tissue, and worsening lung function, thereby leading to a decreased survival rate and poor quality of life [3].

Currently, there are four processes involved in the pathogenesis of RIPF: release of reactive oxygen species (ROS), injury of microvasculature, collection of inflammatory cells, and activation of myofibroblasts [4]. Primary treatment measures in RIPF are mostly supportive, such as supplemental oxygen or pulmonary rehabilitation. Current pharmacotherapies encompassing corticosteroids or azathioprine are proven to be effective only in RP. However, the pharmacological intervention has yet to be determined in RIPF [5]. Studies have revealed that “triple therapy”, including prednisone, azathioprine, and *N*-acetylcysteine, even increases the risk of hospitalization and mortality in patients with RIPF [6]. Pirfenidone and nintedanib are effective treatments for idiopathic pulmonary fibrosis (IPF); however, their role in treatment of RIPF is uncertain [7]. Currently, there are no effective clinical interventions available for RIPF. Thus, new strategies are paramount and urgently needed.

### 1.2. MSCs

Mesenchymal stem cells (MSCs) are multilineage stromal cells with the capability of differentiation and self-renewal. MSCs can be obtained in the umbilical cord, bone marrow, adipose tissue, dental tissue, endometrial polyps, synovial fluid, skin, and placenta from newborns and adults [8,9]. According to the International Society for Cellular Therapy (ISCT) [10], there are basic characteristics defined for MSCs: (1) adherent growth must be observed under standard cultural conditions; (2) surface molecules including CD105, CD73, and CD90 must be expressed in MSCs while the expression of CD45, CD34, CD14 or CD11b, CD79alpha, or CD19 are lacking; (3) MSCs can differentiate into adipocytes, chondroblasts, and osteoblasts in vitro. MSCs also enhance the regeneration of damaged tissues and the differentiation into type II alveolar epithelial cells. In addition, MSCs have low immunogenicity the ability to suppress the release of fibrotic cytokines, and they inhibit epithelial-mesenchymal transition(EMT), a key process in lung fibrosis [11,12,13]. Although MSCs show considerable promise for treatment of acute and chronic inflammatory lung disorders as well as repair of RIPF [14,15,16], the mechanisms are not understood. Herein, we summarize the mechanisms, potential applications, and challenges of MSCs in RIPF.

## 2. The Pathogenesis of RIPF

Radiation exposure induces the generation of numerous reactive oxygen species (ROS), which further causes damage-associated molecular patterns (DAMPs) in the lungs [17]. Inflammatory cells migrate to and accumulate in the injury sites, thereby facilitating the secretion of mediators, such as pro-IL-1β, pro-IL-18, and type I interferon and activation of cell surface-bound TOLL-like receptors (TLR) 2 and 4. Simultaneously, inflammasome NLRP3 can be activated by irradiation leading to upregulation of caspase-1, pro-IL-1β, and pro-IL-18, and the formation of active IL-1β and IL-18. These changes result in cell pyroptosis, a highly inflammatory form of programmed cell death [1,18,19]. Several studies have demonstrated that IL-1β stimulates the release of TGF-β, a key molecule in the process of fibrogenesis in human lung tissue cells. For example, TGF-β/SMAD pathway is associated with lung fibrosis [20]. Specifically, TGF-β binds to transforming growth factor β receptor II (TGFβRII) and leads to phosphorylation of transforming growth factor β receptor I (TGFβRI), thereby increasing phosphorylation of downstream functional molecules, Smad2 and Smad3, the latter usually upregulates the expression of profibrotic molecules [21]. These increased cytokines trigger myofibroblast activation, matrix deposition, and EMT, and eventually leads to lung fibrosis [22] (Figure 1). Ionizing radiation may also induce activation and dysfunction of endothelial cells and then damage the ability of tissue vessels to repair during EMT and tissue fibrosis progression [23]. In addition, the alternative macrophage activation (M2) has been observed rather than classic activation (M1) in the pro-fibrotic immune response [24]. Finally, higher expression of senescence-associated β-galactosidase (SA-β-Gal) and senescence-specific genes (p16, p21, and Bcl-2) are observed in irradiated bone marrow-derived macrophages, suggesting senescent cells may be crucial contributors to RIPF [25,26]. Therefore, multiple cell types are involved in the initiation and development of RIPF.

## 3. Mechanisms of MSC-Based Therapy for RIPF

### 3.1. Homing and Differentiation Processes

Studies have revealed that MSCs have a homing ability; that is, MSCs may migrate to the injured sites and secrete growth factors and chemokines that facilitate tissue repair [27,28]. One study found a 20-fold increase of MSCs in the mouse lungs with thoracic irradiation [29]. A study by Jiang et al. demonstrated that copious transplanted adipose-derived MSC2 may migrate to the injured lung within 12 h after irradiation [30]. The ability of MSC to reach the injured tissues or organs is a prerequisite for their functions.

MSCs homing ability is a complex process associated with a variety of chemokine receptors and their ligands. Stromal cell-derived factor one (SDF-1) upregulated by tissue damage activates the CXCR4—the well-known SDF-1 receptor—then induces migration of MSCs toward injured lung tissue [31], indicating that SDF-1/CXCR4 signal axis plays a crucial role in this process. It has also been demonstrated that the homing of MSC to injured tissues may be regulated by the inflammation state. Among these inflammatory cytokines, TNFα can significantly increase the expression of CXCR4 and enhance the sensitivity of MSC to chemokines [32]. Integrins α4/β1 have been known to participate in the homing process through adhesion to the vascular cell adhesion molecule (VCAM)-1 [33]. Other factors regulating MSCs homing encompass basic fibroblast growth factor (bFGF), matrix metalloproteinases (MMPs), and TLRs [31,34].

Recently, one study by Maria et al. discovered changes in lung-specific Clara and type II pneumocyte cells when MSCs were cocultured with healthy lung tissue [35]. MSC transplantation may increase MSC residency in the lung by activating the Wnt/β-catenin signaling pathway to promote MSC differentiation to ATII, improving lung epithelial permeability and alleviating inflammation in rat models [36]. These data show in response to lung injury, MSCs may reach the damaged area rapidly and differentiate the cell types for improvement of lung function.

### 3.2. Paracrine Effects

MSCs have homing and differentiation abilities. However, their relative percentage in lung is too low to account for their significant therapeutic effects [30]. For example, the replacement of injured sites by differentiated storm cells is only around 5% [37], suggesting that the regenerative effects are mainly through the paracrine mechanisms [38].

As we discussed above, pro-IL-1β, pro-IL-18, TGF-β, and type I interferon are released after radiation and lead to the accumulation of fibroblasts and the development of fibrosis [39]. MSCs paracrine effects may release interleukin 1 receptor antagonist (IL1Ra) and TNF receptor 1 characterized as a competitive inhibitor of IL-1, thereby suppressing their activity [40]. Animal experiments have shown that rats with Ad-MSCs exhibited significantly reduced levels of pro-fibrotic factors, TGF-β1 and α-SMA [41,42]. Dickkop-1 (DKK-1) is a potent antagonist of the Wnt pathway and plays an important role in fibrosis [43] Studies demonstrated that MSCs may induce DKK1 from external sources and then inhibit the induction of EMT in vitro through the Wnt-pathway [44]. Liu revealed that decorin-modified umbilical cord MSCs promoted the release of interferon-γ and inhibited the expression of collagen type III α1 in lung tissues, thereby attenuating fibrosis progression [45]. Recent studies also demonstrated that the anti-fibrotic effects of MSCs on irradiated lungs by stimulating the endogenous secretion of hepatocyte growth factor (HGF) and prostaglandin E2 (PGE2) [46].

Inflammation is a part of the process of pulmonary fibrosis. MSCs possess the anti-inflammation effect by promoting the expressions of the anti-inflammatory factors consisting of IL-1, IL-6, and IL-10, and reducing the expressions of pro-inflammatory factors, such as IL-6 and interferon β [47]. IL-1Ra secreted from MSCs can degrade inflammasomes and inhibit the expression of IL-1β [48]. Studies also revealed that MSCs can diminish NF-kB nuclear transfer and enhance the secretion of HGF, IL-10, and keratinocyte growth factor (KGF) to attenuate the inflammation state in lung tissues [49].

Moreover, the superoxide dismutase (SOD) secreted by MSC is an effective ROS scavenger, which may protect the lungs from reactive oxygen damage and decrease the level of TGF-β and collagen production, thereby reducing intercellular matrix deposition and improving pulmonary fibrosis. A study by Klein et al. indicated that MSCs counteracted vascular damage and endothelial cell loss after radiation through restoring the reduction in SOD1 levels [50]. A study by Chen et al. demonstrated that manganese superoxide dismutase (Mn-SOD) gene-modified (MSCs) can improve the survival of rats and exert an antifibrotic effect [51]. In another experiment, SOD-3 was administered simultaneously with human umbilical cord-derived MSC (hUC-MSC) to treat irradiated mice, and SOD-3 significantly improved the therapeutic effect of hUC-MSC by inhibiting the proliferation of myofibroblasts [52]. These experiments illustrated the importance of SOD in treating fibrosis and suggested the therapeutic potential of MSC in treating RIPF.

### 3.3. Immunomodulatory Effects

It is known that thorax irradiation may induce the recruitment of immune cells to the pulmonary system. For instance, a Th2-like immune response is involved in RIPF [53]. It is also wildly understood that MSCs may regulate intrinsic and adaptive immune cells through direct cell-to-cell contact or production of soluble factors [54,55], and suppress the proliferation of CD4+ T cells, CD8+ T cells, B cells, NK cells, dendritic cells (DCs), and regulatory T cells (Tregs) [56,57,58]. A study by Akiyama et al. suggested that MSCs possessed the ability to promote activated T cells apoptosis through the Fas/Fas ligand pathway [59] while other studies indicated that MSCs may support the survival of T cells in a quiescent state, which is related to the CD95–CD95-ligand-associated cell death [60]. Studies also proposed the mechanism by which MSCs induce the proliferation of T-regs is through release of the human leukocyte antigen-G5 (HLA-G5) [61]. Except for the direct suppression of T cells, MSCs may also curb the generation of Th1, Th2, and Th17 cells by modulating the antigen-presenting function of DCs via IL-6, IL-10, and PGE2 [14]. B cells are the other major population that participate in adaptive immune response in addition to T cells. MSCs may downregulate the expression of CXCR4, CXCR5, and CCR7, thereby leading to a negative modulation in B cell activation [62]. Several signaling pathways, such as extracellular response kinase 1/2 (ERK1/2), PI3K/AKT/mTOR, and p38 are involved in these steps [63]. Moreover, MSCs block the stimulatory activity of DCs to NK cells and result in the impairment of antigen presentation to T cells and the inability of T cells to proliferate [64].

Several immunosuppressive soluble factors, such as indoleamine 2,3-dioxygenase (IDO), nitric oxide (NO), PGE2, heme oxygenase-1 (HO-1), TGF-β, have been reported to be involved in the immunomodulatory process [65]. In the late stage of inflammatory response, the cytokines IL-1β and TNF-α secreted by Th1 and Th2 induce the migration of MSCs to the injury sites with an environment of increased inflammatory cytokines, such as IFN-r and TNF-α. These cytokines participated in the regulation of T cell suppression by MSCs and downregulated the secretion of immune cytokines to reduce lung injury [66]. Therefore, MSCs can mitigate RIPF through immunosuppression and offer an alternative way to regulate the immune response in the treatment of RIPF.

Although the immunosuppressive effect of MSCs has been well clarified, it is important to point out that MSCs may also stimulate the immune system to exert their immunomodulatory function through releasing proinflammatory molecules when the levels of inflammatory cytokines are low [64], suggesting that the specific immunomodulatory effect of MSCs may depend on the inflammatory environment [67].

### 3.4. MSC-Derived Exosome

In addition to the release of a variety of soluble cytokines we discussed above, MSCs may also secret a series of extracellular vesicles (EVs) to exert their function [68]. MSCs-derived exosomes are membranous extracellular vesicles with a lipid bilayer structure that ranges from 30 to 2000 nm in diameter [69]. EVs may present stably in a large number of body fluids, such as breast milk, bronchoalveolar lavage fluid, saliva, blood, and urine [70]. EVs can also mediate messenger RNAs, microRNAs (miRNAs), and proteins to the recipient cells for targeted delivery of genetic information, thereby altering the biological properties of the target cells [71]. Exosomes exhibit unique strengths for biological functions of MSCs, since EVs may travel freely through blood owing to their small size and reach injured sites rapidly and efficiently [72]. Exosomes are considered to be low immunogenicity since they do not express MHC I or II antigens, suggesting a promising cell-free therapeutic strategy [73]. Studies demonstrated that exosome miR-466f-3p derived from MSCs may possess anti-fibrotic effect and preclude radiation-induced EMT through inhibition of AKT/GSK3β [74]. Another study by Lei et al. showed that MSC-EVs may attenuate lung radiation injury through transferring miR-214-3p [75]. Thus, MSCs-exosomes represent a promising therapeutic strategy for treating RIPF.

However, clinical translation may be difficult due to the low amount of EVs production [16]. A widely accepted protocol for exosome isolation, production, and evaluation is still lacking [76].

## 4. Effectiveness of MSCs in Pre-Clinical and Clinical Studies of RIPF

Animal models have been widely used to evaluate the therapeutic effect of MSCs. Most studies have indicated an anti-fibrosis effect of MSCs through detecting histopathological features and cytokines associated with fibrosis. Shao et al. found that adipose-derived MSCs (Ad-MSCs) may inhibit EMT in irradiated type II alveolar epithelial cells and diminish fibroblast activation in mice through the DKK-1/Wnt/β-catenin pathway [44]. MSCs treatment led to an improvement in blood oxygen partial pressure and honeycomb shadows in pulmonary pleurae through CT scan in a canine model [77] and a higher survival rate in a mouse model [40]. In addition to traditional MSCs therapy, genetically and molecularly modified stem cell therapies also show potential. For example, researchers studied umbilical cord-derived MSCs modified with CXCR4, superoxide dismutase 3, or decorin (DCN) and found more better outcomes compared to conventional MSCs in anti-fibrosis function [45,52,78]. These pivotal studies in the last ten years are listed in Table 1.

Kursova reported an improvement in pulmonary fibrosis in 11 patients diagnosed with RIPF treated with MSCs and standard treatment within a one-year follow-up. Even though it was not clear whether it was a drug effect or MSC efficacy, the study still demonstrated the safety of MSCs [79]. A Phase I study (https://clinicaltrials.gov/show/, accessed on 25 November 2022, NCT02277145) included eight patients diagnosed with RIPF and found UC-MSC treatment did not harm the liver, kidney, or other major organs of patients and reduced the clinical symptoms and the density of pulmonary fibrosis. Another Phase I-II study (http://www.chictr.org.cn, accessed on 25 November 2022, ChiCTR1800019309) recruited twelve patients diagnosed with chronic phase radiation-induced pneumonitis or radiation-induced pulmonary fibrosis. The safety was verified with transplanted human umbilical cord MSCs by intravenous infusion (1 × 10^8^/person) once every other week with a total of three times. These clinical trials preliminarily confirmed the safety and effectiveness of MSCs. However, the exact therapeutic effect and mechanism need to be further confirmed by more rigorous studies. Despite all the positive results, it is still noteworthy that a pro-tumorigenic effect was also found is MSCs [80].

**Table 1 cells-12-00006-t001:** Summary of important preclinical studies examining the efficacy of MSC in RIPF.

Year	Model	Dose	Cell Source	Cell Dose	Timepoint after IR	Biological Function	Cytokines/Pathway Involved
2021	BALB/c mice [44]	20Gy	Ad	4 × 10^6^	1 month	inhibited EMT and fibroblast activation	Wnt/β-catenin
2019	C57BL/6 mice [79]	13Gy	CXCR4- UC	5 × 10^5^	1 d	improved histopathological changes	SDF-1, TGF-β1, α-SMA, collagen I
2019	C57BL/6 mice [52]	20Gy	SOD3-UC	1 × 10^6^	2 h	attenuated collagen deposition and myofibroblast proliferation	TGF-β1, IFN-γ, collagen I
2019	Sprague-Dawley rats [42]	15Gy	Ad	5 × 10^6^	2 h/7 d	increased the number of SP-C, inhibited ATII to fibroblastic phenotype	TGF-β1 and α-SMA, TNF-α, IL-1 and IL-6
2018	Beagle dogs [78]	15Gy	UC	1 × 10^6^/kg	180 days	reduced oxidative stress and inflammation	TGF-β-Smad2/3 pathway, TGF-β
2018	C57BL/6 mice [45]	20Gy	decorin (DCN)-modified-MSC	1 × 10^6^	6 h/28 days	alleviated histopathologic injuries and later fibrosis	IFN-γ, Tregs
2017	NOD/SCID mice [51]	13Gy	MnSOD- MSC	1 × 10^6^	1 day	improved survival anti-fibrotic	TGF-β1
2017	C57BL/6 Mice [50]	15Gy	AO/BM	0.5 × 10^6^	24 h/14 d	counteracted radiation-induced vascular damage and EC loss	SOD1, Mmp2, Ccl2, Plau/uPA
2016	C57BL/6 mice [40]	18Gy	BM	1 × 10^3^ /5 × 10^3^ /1 × 10^4^	24 h	improved survival and histopathological features	SPC, PECAM, IL-10, TNF-α
2015	Sprague-Dawley rats [30]	15Gy	Ad	5 × 10^6^	2 h	Anti-inflammation anti-fibrotic-maintained lung epithelium integrity	IL-1, IL-6, IL-10, TNF-α, TGF-β1, CTGF, α-SMA, collagen
2015	Sprague-Dawley rats [46]	15Gy	Ad	5 × 10^6^	NM	Inhibited EMT	HGF, PGE2, TNF-α, TGF-β1
2013	C57BL/6 mice [29]	NM	Ad-sTβR-MSC	NM	day 0/day 14	alleviated survival and histopathology data	MDA, CTGF, α-SMA
2013	C57BL/6 mice [81]	20Gy	HGF- MSC	1 × 10^6^	6 h	improved histopathological and biochemical markers of lung injury	TNF-α, IFN-γ, IL-6, ICAM-1

NM: not mentioned IR: ionizing radiation; EMT: epithelial-mesenchymal transition; Ad-MSCs: adipose-derived mesenchymal stromal cells; UC: umbilical cord blood; Ao: aorta; DKK-1: Dickkop-1; SDF-1: stromal cell-derived factor-1; EC: endothelial cell; α-SMA: alpha-smooth muscle actin; HGF: hepatocyte growth factor; PGE2: prostaglandin E2; SP-C: lung epithelial cells; SOD: superoxide dismutase; MnSOD: superoxide dismutase; ICAM-1:intercellular adhesion molecule-1; Mmp2: matrix metalloproteinase 2; Ccl2: chemokine (C-C motif) ligand 2; Plau/uPA: urokinase-type plasminogen activator; NOD/SCID: nonobese diabetic/severe combined immunodeficiency.

Many factors may influence the efficacy of MSCs. First, culture conditions are associated with the phenotype and function of MSCs. Therefore, cells with superior functionality can be obtained by modulating culture conditions. While MSCs are normally cultured with 20% oxygen tension, studies revealed that hypoxic conditions may reduce MSC senescence, increase proliferation, and significantly enhance the immunosuppressive ability to suppress T-cell proliferation [82]. Lan et al. demonstrated that hypoxia-preconditioned MSCs attenuate bleomycin-induced pulmonary fibrosis through and increase in the production of anti-inflammatory and anti-fibrotic cytokines [83]. H_2_O_2_-preconditioned MSCs in mice with bleomycin-induced pulmonary fibrosis results in significantly decreased connective tissue and collagen deposition compared to untreated cells [84]. In addition to modulating culture conditions, genetically modified MSCs exhibit a more efficient therapeutic effect. For example, hepatocyte growth factor (HGF)-modified Ad-HGF-MSC showed a better ability to reduce expression of inflammatory cytokines, thereby protecting ATII cells and inhibiting pulmonary fibrosis [81]. To maximize the therapeutic benefits of MSCs, in-depth studies related to the optimal timing of MSCs administration have been conducted. MSCs injected 4 h after thoracic exposure reach the damaged sites and differentiate into functional cells. This leads to increased deposition of collagenous fiber after 60 and 120 days, suggesting the importance of the time window in the treatment of RIPF [85]. Jiang described a strategy with two Ad-MSCs vein injections, one at 2 h and another at 7 days, were effective in abating lung fibrosis compared with single delivery at either time point [42]. In addition, different types of MSCs may differ in treatment efficacy. Bone marrow-derived MSCs are the most commonly used source, while adipose tissues are more accessible for the generation of MSCs. The number of transplanted cells also needs to be studied. In a mouse model of radiation damage treated with BM-MSC, the therapeutic effects of high (1 × 10^4^ cells/g), medium (5 × 10^3^ cells/g), and low (1 × 10^3^ cells/g) doses of transplanted cells were compared, and the results showed that low doses of cells had better results for lung damage [40].

## 5. Conclusions

RIPF is a fatal adverse effect of radiotherapy for thoracic tumors and there is no effective pharmacological therapy available. MSCs with their copious sources, bioactive characteristics, easy cultivation, and low immunogenicity have unique advantages in the treatment of RIPF due to their homing, differentiation, paracrine, and immunomodulatory effects. Genetically and molecularly modified stem cell therapies have also shown potential in clinics. Several studies also suggested MSC-derived exosomes as a promising treatment for RIPF. While abundant preclinical studies have demonstrated the effectiveness and safety of MSCs in treating RIPF, more clinical trials are still needed. Current clinical studies demonstrate safety whereas the validity in patients requires more evidence. Moreover, further studies are needed to identify the ideal culture conditions, time point of application, optimal cell source, and dose in MSCs treatment.

## Figures and Tables

**Figure 1 cells-12-00006-f001:**
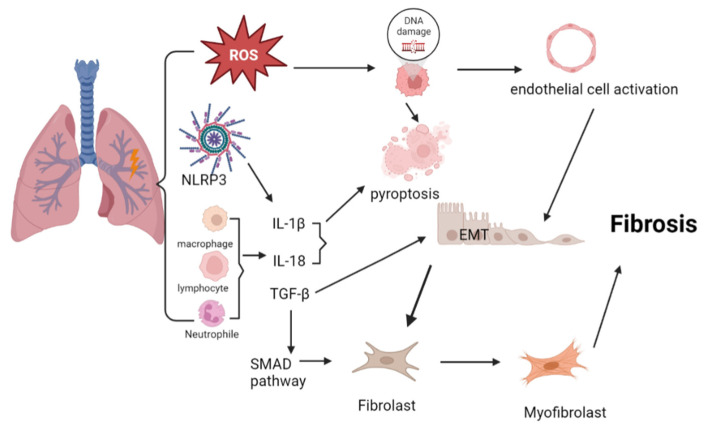
The mechanisms of RIPF: irradiation causes the generation of ROS, which results in DNA damage and activation of the endothelial cells. Inflammasome and immune systems are activated, thereby releasing IL-1β and IL-18 and leading to pyroptosis. IL-1β promotes the release of TGF-β, which triggers TGF—β/SMAD signaling pathway and induces EMT and fibrosis. (This figure was made with Biorender.).

## Data Availability

Data sharing not applicable.

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
