# Peer review of "Mesenchymal Stem Cells in Radiation-Induced Pulmonary Fibrosis: Future Prospects"

_cells, 2022, doi:10.3390/cells12010006_

Round 1

Reviewer 1 Report

First of all, we would like to thank the authors for their efforts in carrying out this review. The use of mesenchymal stem cells to treat radiation-induced pulmonary fibrosis is an interesting emerging treatment option.

However, there are a number of comments that I believe would help to complete the review:

-          - The article should be better structured. The section titles lack continuity. For example, item 2 is entitled: Progresses in mechanisms of MSC-based therapy for RIPF, and number 3: Immunomodulatory effects. Clearly point 3 is one of the mechanisms of action of MSCs that should be included as part of number 2. A common thread is missing in the article, which is currently just an accumulation of interesting quotes.

-          - In Table I, a large number of relevant papers are included, however, the "Model" column is uninformative. They should include the strain of rat/mouse/dog used.

-          - There are recent articles on the subject that have not been included. Just as an example: Zanoni M, Cortesi M, Zamagni A, Tesei A. The Role of Mesenchymal Stem Cells in Radiation-Induced Lung Fibrosis. Int J Mol Sci. 2019 Aug 8;20(16):3876. doi: 10.3390/ijms20163876. PMID: 31398940; PMCID: PMC6719901.

- MSCs have already been used as a therapeutic agent to treat radiation-induced pulmonary fibrosis in clinical trials. The authors should search clinicaltrials.org, reference the studies, and discuss their findings.

Author Response

Comment 1: The article should be better structured. The section titles lack continuity. For example, item 2 is entitled: Progresses in mechanisms of MSC-based therapy for RIPF, and number 3: Immunomodulatory effects. Clearly point 3 is one of the mechanisms of action of MSCs that should be included as part of number 2. 

Response: Many Thanks. We reorganized the manuscript into five sessions: 1. Introduction, 2. The pathogenesis of RIPF, 3. Progresses in mechanisms of MSC-based therapy for RIPF (Homing and differentiation processes, Paracrine Effects, Immunomodulatory Effects, MSC-derived exosomes), 4. MSCs in pre-clinical and clinical studies of RIPF, 5. Conclusion.

Comment 2: In Table I, a large number of relevant papers are included, however, the "Model" column is uninformative. They should include the strain of rat/mouse/dog used.

Response: Thanks for your valuable suggestion. We have added the strain of rat/mouse/dog used in Table 1 to make it more informative.

Comment 3: There are recent articles on the subject that have not been included. Just as an example: Zanoni M, Cortesi M, Zamagni A, Tesei A. The Role of Mesenchymal Stem Cells in Radiation-Induced Lung Fibrosis. Int J Mol Sci. 2019 Aug 8;20(16):3876.DOI:10.3390/ijms20163876.

Response: Thanks for your suggestion. We searched the latest literature and added this article you refred and another one (DOI: 10.3390/cells10020294) in the references.

Comment 4: MSCs have already been used as a therapeutic agent to treat radiation-induced pulmonary fibrosis in clinical trials. The authors should search clinicaltrials.org, reference the studies, and discuss their findings.

Response: Thanks for your suggestion. We searched for clinical studies reported in https://clinicaltrials.gov database using “Mesenchymal Stem Cells” and “Pulmonary Fibrosis” as keywords and found there was a Phase I+Phase II study (ChiCTR1800019309) and a Phase I study (NCT02277145) on the clinical use of MSCs in RIPF. We discussed their findings in the revised paper.

Reviewer 2 Report

Brief summary : In the review entitled “Mesenchymal stem cells in radiation-induced pulmonary fibrosis: looking forward”, Chen et al. summarize the knowledge on the mechanisms and use of mesenchymal stem cells (MSC) to treat radiation induced pulmonary fibrosis (RIPF). Briefly, they first define RIPF, MSC and list the known molecular RIPF determinants. Then, the authors introduce the different studies providing mechanistic evidence for RIPF treatment using MSC. Lastly, they describe the immunomodulatory effects of MSC and this last section also include information on the MSC-derived exosome as well as the preclinical data supporting the use of MSC to treat patients presenting RIPF. This manuscript will provide the reader interested in MSC therapeutic strategies the information on molecular and cellular mechanisms as well as preclinical data supporting the use of MSC to treat patients with RIPF.

General concept comments : RIPF is still a deadly disease induced by radiotherapy and new therapeutic options such as MSC are needed. The manuscript is written correctly but I would suggest to have it read by a native English speaker in order to correct grammar and typos. Regarding the structure, I would suggest to include the immunomodulatory effects and exosomes paragraphs in the “Mechanisms of MSC-based therapy for RIPF” section and have a dedicated section on the preclinical and clinical data supporting the use of MSC in patients.

Specific comments :

·        - The authors introduce the term RIPF and then switch to RILF. To facilitate reader’s understanding, I would suggest to stick to one single abbreviation.

·        - On the pathogenesis of RIPF, I would include the contribution of vascular system (see the review from Baselet et al, https://doi.org/10.1007/s00018-018-2956-z) and also, in Figure 1, I would add the activation of endothelial cells and EndoMT that participate to fibrosis progression.

·        - Also in Figure 1, the link between inflammasome, DNA damages and pyroptosis is not clear. If made with Biorender, it should be cited in the legend.

·        - I would remove the term multipotent (L43) that is redundant with the differentiation and self-renewal capacity.

Author Response

Comment 1: The authors introduce the term RIPF and then switch to RILF. To facilitate reader’s understanding, I would suggest to stick to one single abbreviation.

Response: Thanks. We revised our paper sticking to RIPF and deleted the term RILF.

Comment 2: On the pathogenesis of RIPF, I would include the contribution of vascular system (see the review from Baselet et al, https://doi.org/10.1007/s00018-018-2956-z) and also, in Figure 1, I would add the activation of endothelial cells and EndoMT that participate to fibrosis progression. 

Response: Thanks for your suggestion. We added this in Figure 1 as you suggested.

Comment 3: Also in Figure 1, the link between inflammasome, DNA damages and pyroptosis is not clear. If made with Biorender, it should be cited in the legend.

Response: Thanks. We redrew the figure with Biorender in the revised paper to clarify that generation of ROS resulted in DNA damage

.

Comment 4: I would remove the term multipotent (L43) that is redundant with the differentiation and self-renewal capacity.

Response: Thanks. We removed the term multipotent in our revised paper.